

**Assessing the effect of flood restoration on surface-subsurface interactions in**
**Rohrschollen Island (Upper Rhine River – France) using integrated hydrological**
**modeling and thermal infrared imaging**
Benjamin JEANNOT[1], Sylvain WEILL[1*], David ESCHBACH[2,3], Laurent SCHMITT[2]
and Frederick DELAY[1]
[1] *Université de Strasbourg, CNRS, ENGEES, LHyGeS UMR 7517, F-67000 Strasbourg, France*
[2] *Université de Strasbourg, CNRS, ENGEES, LIVE UMR 7362, LTSER - "Zone Atelier*
*Environnementale Urbaine", F-67083 Strasbourg, France*
[3] *Sorbonne Université, CNRS, EPHE, UMR 7619 Metis, 75005 Paris, France*
* Corresponding author
Laboratoire d'Hydrologie et de Géochimie de Strasbourg
Université de Strasbourg - CNRS - UMR 7517
1 rue Blessig, 67000 Strasbourg, France
Tél : +00 33 3 68 85 03 86
Mail : s.weill@unistra.fr



**Abstract**
Rohrschollen Island is an artificial island of the large Upper Rhine River whose geometry and
hydrological dynamics are the result of engineering works during the 19$^{th}$ and 20$^{th}$ centuries.
Before its channelization, the Rhine River was characterized by an intense hydro-
morphological activity which maintained a high level of biodiversity along the fluvial
corridor. This functionality considerably decreased during the two last centuries. Since 2012,
a restoration project was launched to reactivate typical alluvial processes, including bedload
transport, lateral channel dynamics and surface-subsurface water exchanges. An integrated
hydrological model has been applied to the area of Rohrschollen Island to assess the
efficiency of the restoration regarding surface and subsurface flows. This model is calibrated
using measured piezometric heads. Simulated patterns of water exchanges between the
surface and subsurface compartments of the Island are checked against the information
derived from thermal infrared imaging. The simulated results are then used to better
understand the evolutions of the infiltration/exfiltration zones over time and space and to
determine the physical controls of surface-subsurface interactions on the hydrographic
network of Rohrschollen Island. The use of integrated hydrological modeling has proven to be
an efficient approach to assess the efficiency of restoration actions regarding surface and
subsurface flows.

**Keywords**
Surface-subsurface water interactions, Integrated hydrological modeling, flood restoration,
Thermal infrared imagery, Rohrschollen Island, Upper Rhine River.

**Highlights** (less than 85 characters, including spaces)

-   Direct hydrological impacts of restoration on a riverine island are modeled.



-    Integrated modeling captures the hydrologic surface-subsurface interactions.
-    Simulated exfiltration areas are also located by thermal infrared imaging.
-    Management practices can be optimized on the basis of simulated system responses.

**1. Introduction**

Interactions between surface and subsurface flow processes are key components of the

continental hydrological cycle (Winter, 1995; Sophocleous, 2002), which have received
particular attention in the last decades partly because of their substantial impact on the overall
response of hydrologic systems (Boano et al., 2014; Brunner et al., 2017, and citations
herein). Several studies have recently highlighted the hydrological interactions between
surface and subsurface that have a major impact on the biogeochemical and ecological
responses of hydrosystems (e.g., Stegen et al., 2016; Danczak et al., 2016; Partington et al.,
2017; Stegen et al., 2018). These interactions, which are partly driven by the
geomorphological structure and the channel dynamics (Namour et al., 2015), influence flow
pathways, water mixing, residence time in the hyporheic zone along streambeds, and the
overall ecological functioning (Schmitt et al., 2011). They are complex for several reasons,
including (a) the nonlinearity of the processes involved, (b) the strong heterogeneity of the
hydrological systems, and (c) the incidence of small-scale features on large-scale behavior
(Hester et al., 2017). Although these surface-subsurface interactions have been extensively
investigated in the last decades, several issues relating to them require a deeper understanding
to address contemporary challenges associated with water quality and water resources
management (Brunner et al., 2017). Among these issues, monitoring and modeling the
evolution of these interactions over space and time is fundamental (Krause et al., 2014),
especially in the context of river restoration.





River restoration has been applied worldwide to counteract the undesired effects of
anthropogenic actions on river ecosystems and ecosystem services (e.g., Wohl et al., 2015,
and citations herein). From a general perspective, the goal of restoration projects is to enhance
the hydrological, biogeochemical, and ecological functioning of large rivers and stream
hydrosystems through the reactivation of lost geophysical, geochemical, or biological
processes. Due to their firm control on biogeochemical and ecological signatures in the so-
called hyporheic zone (e.g., Perralta-Maraver et al., 2018), the interactions between surface
and subsurface hydrological processes may become a focus of restoration projects (e.g.,
Boulton et al., 2010; Friberg et al., 2017). Many projects try to improve the water quality
and/or ecological processes of the hydrosystem through engineering works that target
hyporheic exchange enhancements. Maintaining or amplifying these interactions could reveal
crucial regarding climate change effects to preserve aquatic species. Nevertheless, it is still
very difficult to assess the efficiency of such restoration projects as this requires a refined
characterization of the location and amplitude of surface-subsurface interactions (e.g.,
Morandi et al., 2014).
Several advances in measurement techniques and modeling approaches appear very
promising to improve our current understanding and our forecasting capabilities regarding
surface-subsurface interactions (Krause et al., 2014; Brunner et al., 2017). Many
experimental/field projects are related to the use of temperature as a tracer of hydrological
connectivity and locations where groundwater discharges into surface water bodies (e.g.,
Pfister et al., 2010; Daniluk et al., 2013). Two different thermal techniques—Fiber Optic-
Distributed Temperature Sensing (FO-DTS) and Thermal InfraRed (TIR) survey—have been
used for their potential to inform on spatial and temporal patterns of water fluxes in large
areas of the hyporheic zone through the determination of thermal anomalies. FO-TDS
provides one-dimensional profiles of these anomalies with a fine spatial resolution by





submerging fiber optic cables along a streambed. TIR survey can be performed from air and
satellite, and informs on surface temperature with two-dimensional images of various
resolutions (e.g., Hare et al., 2015).

For their part, integrated hydrologic models emerged in the late 1990s, and they are

now recognized as suitable tools to investigate streamflow generation processes at the
catchment scale (e.g., Paniconi and Putti, 2015; Fattichi et al., 2016). Although most
integrated models rely on the solution to the 3-D Richards equation to describe subsurface
flow (e.g., Maxwell et al., 2014), alternative low-dimensional approaches that simplify the
description of the subsurface compartment (still with some physical meaning) have recently
appeared (e.g., Hazenberg et al., 2015, 2016; Jeannot et al., 2018). Solving the 3-D Richards
equation with a proper discretization to capture the complex and small-scale physics of flow
in the vadose zone over large areas may require substantial computer resources. Low-
dimensional integrated approaches that are efficient regarding computation time could also
reveal beneficial to tackle practical water management issues. Integrated models, irrespective
of their level of complexity, explicitly account for the interaction between surface and
subsurface hydrological processes. Thus, their application to hydrosystems renders insights on
the evolution over time and space of surface-subsurface interactions (e.g., Partington et al.,
2013; Camporese et al., 2014).

Hydrologic modeling has already been used to assess the potential effects of

restoration works on the hydrologic response of a given system (e.g., Martinez et al., 2014;
Ohara et al., 2014; Clilverd et al., 2016). The studies reported in the ongoing literature mostly
deal with the effect of restoration on water table dynamics, flood frequency, ecosystem
services (e.g., thermal refuges for specific species), and vegetation dynamics. To our
knowledge, the prediction with models of hyporheic exchanges has not yet been considered.
No integrated hydrologic model has been applied to a restored fluvial hydrosystem even





though the application could reveal noteworthy data in rendering quantitative indicators of
restoration efficiency. In addition, the combined use of thermal information with integrated
hydrological models is not yet common even though comparing and discussing both seems
fruitful. Ala-aho et al. (2015) used thermal imaging and integrated modeling to study the
exchanges between groundwater and lakes in Finland. Glaser et al. (2016) used integrated
modeling and TIR survey to improve the calibration procedure and investigate the dynamics
of the saturated area in a small catchment in Luxembourg. Munz et al. (2017) combined
thermal measurement along the banks of a stream and integrated modeling at the reach scale
to improve the determination of residence times in the hyporheic zone.

This paper aims to present how an integrated hydrologic model was used in

combination with data of TIR imaging to specifically investigate the effect of a restoration
project on surface-subsurface water interactions.  A main goal, which is also an innovation, is
to propose a method that quantitatively evaluates the efficiency of restoration actions
regarding the enhancement of hyporheic exchanges and their dynamics over time and space.

**2. Data and hydrological modeling**
*2.1. Study Area – Rohrschollen Island*
*2.1.1 General description*

Rohrschollen Island is an artificial island located 8 km south of Strasbourg (Upper

Rhine, France, see Fig. 1-a), as the result of historical engineering works carried out along the
Rhine River mainly to prevent flooding and to develop navigation and agriculture. The
hydrological and geomorphological dynamics of the area were massively impacted (Eschbach
et al., 2017; Eschbach et al., 2018). Three structures completely control the current geometry
and hydraulic behavior of Rohrschollen Island (Fig. 1): (a) the diversion dam (built in 1970)
at the southern end of the island that diverts most of the river flow into the Rhine Canal at the



western bank of the island, (b) the hydropower plant (built in 1970) located on the Rhine
Canal downstream to Rohrschollen Island, and (3) an agricultural dam (built in 1984) at the
northern part of the Island to keep a constant water level in the by-passed Old Rhine at the
eastern bank of Rohrschollen Island.

Rohrschollen Island was regularly flooded in the past (Eschbach et al., 2018). The

main anastomosed channel inside the Island, the Bauerngrundwasser (BGW; Fig. 1), was
disconnected on its upstream mouth from the Rhine River by the excavation of the Rhine
canal. This disconnection, combined with dampened groundwater dynamics along the Island,
degraded the hydrological, geomorphological, and ecological functioning of the hydrosystem.
The former flood dynamics induced large water table fluctuations, lively interactions between
the surface and subsurface domains, intense rejuvenation of habitat mosaic driven by
geomorphological processes, and a high level of biodiversity for species of aquatic and
riverine habitats. As a result of engineering works performed to control the Rhine River, the
ecological services associated with the flood dynamics and the hydrologic connection
between the floodplain of the Island and the river were lost.

In 2012, the European Union funded a restoration project (LIFE + program) in order to

counteract the loss of various natural processes and thus re-establish part of the former
dynamics of the system. The Rhine River water is now injected through a floodgate into a 900
m long new artificial channel (south of the Island; Fig. 1-b) when the upstream discharge in
the Rhine River exceeds 1550 $m^3s^{-1}$. These injections should contribute to (a) enhancing
discharge into the surface water bodies of the Island (especially in the BGW) and partly
recovering floods on the Island (floods occur when the injected rate exceeds the top-edge
discharge of the new channel at 20 $m^3s^{-1}$), (b) recovering bedload transport and lateral channel
dynamics (especially along the new channel), (c) activating surface-subsurface interactions,



and (d) stimulating the renewal of aquatic and riverine ecosystems. The injected discharges
range between 2 and 80 m$^3$ s$^{-1}$, according to the total discharge in the Rhine River.
*2.1.2 Hydrologic monitoring*

A large interdisciplinary environmental monitoring was conducted to investigate the

effects and the efficiency of the restoration, but also to check on some risks such as the
eventual collapsing of the new channel banks under strong water injections. As an example, a
dense network of piezometers (yellow squares in Fig. 1-c) was installed along both the
artificial new channel and the BGW. More precisely, ten transects along these channels were
instrumented with a piezometer on each channel bank. The time resolution of measurements
in the 20 piezometers ranges from 5 min along the new channel to 10 min along the BGW.
This network is particularly crucial for hydrological model calibration and to understand the
interactions between groundwater and surface water bodies. Other subsurface head
measurements are also available on the eastern and western sides of the island. The French
National Electricity Company (EDF) is operating devices at the western side of the Island
(along the Rhine Canal) to monitor the state of the dike road (blue squares in Fig. 1-c) and, as
the owner and manager of the Rohrschollen Island Nature Reserve, the city of Strasbourg is
following subsurface water table dynamics at the eastern side (orange squares in Fig. 1-c).

*2.1.3 Historical and sedimentological surveys*

Geo-historical and sedimentological surveys were used to reconstruct the morpho-

sedimentary temporal trajectory of the Island since the middle of the 18$^{th}$ century. The geo-
historical survey is partly based on six old maps, two sets of aerial photographs, and the actual
digital elevation model of the Island (see Fig. 2, left part). Planimetric data were
georeferenced in a GIS (geographic information system) and processed to highlight the
temporal dynamics of the main morpho-ecological units. The sedimentological study was





based on seven coring transects distributed along the BGW. Grain size analysis was also
performed on sediment samples from three transects and two pits in the floodplain to
determine the transport and deposition processes of fine sediments. The combination of the
geo-historical and sedimentological analysis helped to reconstruct the sedimentary deposition
trajectory and to locate precisely historical gravel bars (see Fig. 2, right side). This
information was used to spatialize the parameters of the hydrological model and to preset the
initial values of key parameters related to the composition of the sediment units. More details
on this part of the study can be found in Eschbach et al. (2018).

*2.1.4. Thermal infrared imaging*

Thermal infrared imaging (TIR) was carried out at Rohrschollen Island to investigate

the relationship between the evolution of some geomorphological features (e.g., riffles and
pools) and the interactions between surface and subsurface waters. A FLIR b425 infrared
camera was fixed under a paraglider to take pictures covering the whole island. The camera
was calibrated using several key parameters such as water emissivity and the height above the
topography. The flight took place on January 22, 2015, a date chosen to have minimal canopy
extension and maximal temperature contrast between surface and subsurface waters (with
approximately 4°C surface temperature and 10°C groundwater temperature). The thermal
images were processed to locate thermal anomalies along the new artificial channel and the
BGW. The radiance was first converted into temperature using Planck's law and in-situ
measurements as references. The temperature maps were then georeferenced, and pixels
associated with high uncertainty on temperatures were also discarded. Further treatments
based on optic images (in the visible wavelengths) delineated and located surface objects such
as banks, vegetation, logjams, and gravel bars. Further details about thermal image processing
can be found in Eschbach et al. (2017).



*2.2. Hydrological modeling strategy*

*2.2.1. The Normally Integrated Model (NIM)*

The integrated hydrological model used to model Rohrschollen Island is the Normally Integrated Model (NIM) (Pan et al., 2015; Weill et al., 2017; Jeannot et al., 2018). This tool is a physically based and spatially fully-distributed model that describes flow processes in the surface and subsurface domains of a catchment and their couplings. For the sake of simplicity, only the model parts used for this study are presented here. A detailed presentation of the model (primarily concerning treatment of the flow equations) is available, for example, in Jeannot et al. (2018).

The subsurface flow processes are described using a low-dimensional equation that results from the integration of the 3-D Richards equation along a direction normal to the bedrock (i.e., the impervious bottom of the aquifer). The final equation for subsurface flow can be written as:

$$\frac{\partial \bar{\theta}}{\partial t} + \bar{S}(h)\frac{\partial h}{\partial t} + \nabla_{x,y}\cdot\left(-\bar{\mathbf{T}}(\theta)\nabla_{x,y}h\right) = Q_w \qquad [1]$$

where $\bar{\theta}(h) = \int_{z_w}^{z_s}\theta(z)dz$, $\bar{S}(h) = S_{sat}\,h$, $\bar{\mathbf{T}}(h,\theta) = \mathbf{K_{sat}}\,h + \int_{z_w}^{z_s}\mathbf{K}\left(\theta(z)\right)dz$. $\mathbf{K_{sat}}$, and $S_{sat}$ are averages along the integration direction $z$ of the saturated hydraulic conductivity tensor and the specific storage capacity in the saturated zone, respectively. $\theta$ [-] is the water content; $\mathbf{K}$ [LT$^{-1}$] is the tensor of hydraulic conductivity; $h$ [L] is the hydraulic head (or the capillary head); and $Q_w$ [LT$^{-1}$] is a source term that accounts for the subsurface interactions with both the 1-D river network and the 2-D overland flow. It is worth noting that the 1-D river network compartment was not used in this study because the precision of the digital elevation model (Fig. 2, left) was enough to delineate and model streams, channels, and other small water routing in slight topographic depressions of the 2-D overland flow layer.



The 2-D overland flow layer is described using the so-called diffusive wave equation,
which is written as:
$$\frac{\partial (h_s + z_s)}{\partial t} - \nabla_x \cdot \left( T_{s,x} \nabla_x \left( h_s + z_s \right) \right) - \nabla_y \cdot \left( T_{s,x} \nabla_y \left( h_s + z_s \right) \right) = q$$    [2]
with

$$T_{s,x} = \frac{h_s^{5/3}}{N_{man,x}^2 \beta \nabla \left( h_s + z_s \right)} \quad ; \quad T_{s,y} = \frac{h_s^{5/3}}{N_{man,y}^2 \beta \nabla \left( h_s + z_s \right)}$$


$$\beta \nabla \left( h_s + z_s \right) = \left[ \left( \frac{\partial \left( h_s + z_s \right)}{\partial x} \right)^2 \frac{1}{N_{man,x}^4} + \left( \frac{\partial \left( h_s + z_s \right)}{\partial y} \right)^2 \frac{1}{N_{man,y}^4} \right]^{1/4}$$

$h_s$ [L] is the water depth at the surface; $z_s$ [L] is the soil surface elevation; $u_x$ and $u_y$ [LT$^{-1}$]
are the water velocity components along the $x$ and $y$ directions (that are locally defined in the
plane normal to the direction of integration $z$ of Eq. (1)); $q$ [LT$^{-1}$] is a source term including
the exchanges with the 1-D river flow compartment and with the subsurface; and $N_{man,x}$ and
$N_{man,y}$ [L$^{-1/3}$T] are the Manning coefficients in the $x$ and $y$ directions, respectively.
The coupling between Eq. (1) and Eq. (2) relies upon a first order law stating that the
flux exchanged between surface and subsurface flows is proportional to the head gradient
between the two compartments. The exchanged flux $Q_{Ex,2D \leftrightarrow SS}$ [LT$^{-1}$] can be formalized as:
$$Q_{Ex,2D \leftrightarrow SS} = K_{Int} \frac{(z_s + h_s) - h}{l_e} F_s$$    [3]
$$F_s = \min \left[ \left( \frac{h_s}{h_{ob}} \right)^{2\left(1 - h_s / h_{ob}\right)} ; 1 \right]$$    [4]
where $K_{Int}$ [LT$^{-1}$] is the vertical hydraulic conductivity at the interface between the surface
and subsurface compartments; $l_e$ is a user-defined coupling length (i.e., an empirical
thickness of the interface between surface and subsurface compartments); $F_s$ [-] is a scaling



function accounting for the saturated-unsaturated character of the interface between the
surface and subsurface; and $h_{ob}$ is the total obstruction height accounting for small
irregularities of the topography.
Regarding the numerical solution, both equations are solved together in a fully implicit
manner using advanced numerical schemes. Note that both equations are two-dimensional and
that only one computation mesh mimicking the topographic surface of the system is required
for simulating both surface and subsurface processes, including their interactions.

*2.2.2 Model setup and parametrization*
The computation mesh for all the simulations of the study was built from data from an
airborne LIDAR survey performed in 2015 that produced high-resolution images of the
topography (50 cm in the horizontal plane and 1-2 cm in elevation). The whole Rohrschollen
Island is meshed using triangular elements of 20 m on a side. The exception is a 120 m wide
corridor surrounding the new channel and the BGW where a refined spatial resolution of 10 m
is used. The higher resolution is assumed to better capture the hydrological dynamics and the
surface-subsurface interactions along the surface water bodies of the Island. Prescribed-head
(Dirichlet) boundary conditions are imposed at the western and eastern banks of Rohrschollen
Island for the subsurface model, and they have been documented by measurements collected
by the EDF and the city of Strasbourg. These boundary conditions may vary over time,
depending on the modeled period and availability of data. The northern and southern parts of
the Island were considered as no-flow boundaries. The initial conditions were set up by
running the model with consistent boundary conditions for the subsurface and an injection
rate of 2 m$^3$ s$^{-1}$ (which is the routine injection rate) at the new channel inlet until stable
hydrological conditions were reached.



The initial parametrization of the model, especially the spatial distribution of
hydrodynamic parameter values, mainly relies upon patterns drawn from the geo-historical
and sedimentological surveys of the island (Eschbach et al., 2018). As an example, Fig. 2
maps three historical snapshots of the main geomorphological units (gravel bars). Results
from particle size analysis also helped to predefine variation ranges of crucial parameters,
such as the hydraulic conductivity and retention curve parameters of the sediments and the
exchange coefficient between surface and subsurface. The spatial distribution of parameters
as various zones (of uniform values within a zone) was directly delineated by relying upon
maps and local information gathered, rendered compatible, and processed via a GIS. Note that
corridors around the new channel, the BGW, and the network of paleo-channels visible in the
floodplain, which the digital elevation model in Fig. 2 identifies, were defined and
parametrized separately to account for specific deposition histories resulting in specific
sediment grain size.

*2.2.3. Model calibration and validation*
The integrated model was calibrated and validated using two periods of time for which
high-rate injections in the new artificial channel were carried out. The first period (December
9, 2014–December 15, 2014) was used as a model calibration exercise which encompassed
two peaks of injection with one reaching 80 m$^3$ s$^{-1}$. The second selected period (May 15,
2015–May 21, 2015) was employed as a validation exercise with three injection peaks, two of
them exceeding 70 m$^3$ s$^{-1}$. Fig. 3 reports the evolution of the injected flow rates over time at
the system inlet for both the calibration and validation periods.
After a first simulation employing the initial parametrization (defined in Section 2.2.2),
manual calibration was carried out for the first period to improve the fitting between
measured and simulated head levels in the subsurface. Both the Root Mean Square Error



(RMSE) and the Kling-Gupta Efficiency (KGE) associated with observed heads at the 10
transects cross cutting the new channel and the BGW were used as indicators to evaluate the
quality of the simulations. Only the hydraulic conductivity and the exchange coefficient
between surface and subsurface were slightly adjusted while trying to preserve the initial
spatial zonation. Fig. 4 maps the final set of parameters for the saturated hydraulic
conductivity and the exchange coefficient, which are the most sensitive parameters of the
model in the present case. The sets of calibrated parameters were then used for simulating the
validation period to check whether the calculated subsurface head levels match up to the
measured values.
It is worth noting here that the calibration exercise was performed over a period where
the TIR images were not available, which means, in turn, that the calibration only relied upon
measured groundwater head levels as a reference. The goal of the calibration was not to match
the exfiltration patterns identified through the TIR imaging. When this information became
available, the simulation period used for the calibration was extended to reach the date of the
airborne flight (January 22, 2015), and the boundary conditions were updated. The exfiltration
patterns were then used as verification information to confirm that the model could properly
describe the interactions between surface and subsurface and thus be used as a forecasting
tool. Forecasts discussed hereinafter cover optimizations of injections in the artificial channel
upstream to the Island, which are mainly supposed to maintain active ponding and wetlands
(mainly from groundwater outcrops) over long periods.

**3. Results and discussion**
*3.1. Model outputs*
Fig. 5 displays the evolution over time of simulated and observed piezometric heads at
two locations (transects) in the island. It also plots simulated versus observed heads for all





locations and sampling times used during the calibration period. Heads at transects in Fig. 5
were selected to show the best and worst match concerning RMSE between simulation and
observation. In general, the model adequately reproduces the system dynamics, capturing the
two peaks of head response associated with the injection patterns at the new channel inlet.
The recession part of the response is also captured well with a slight overestimation of the
final head value for transect T8 (Fig. 5, upper left panel). The plot of simulated versus
observed heads (Fig. 5, right) confirms that the model tends to overestimate the piezometric
heads as more points are located above the 1:1 straight line. This feature is associated with
one of the founding assumptions of the model regarding the vadose zone, which is integrated
with the saturated zone and can be excessively or not sufficiently capacitive, depending on the
mean soil moisture (see Weill et al., 2017). The values of the two performance indicators that
are the RMSE and the KGE are satisfying, at 17 cm and 0.93, respectively.

Fig. 6 depicts the same information as Fig. 5 but for the validation period. The

agreement between simulated and measured heads remains good with an RMSE of 24 cm and
a KGE of 0.75. As is often the case, the results degrade when passing from calibration to
validation exercises. That being said, both exercises show that the NIM and its calibrated set
of parameters render convincing simulations of the highly transient hydrologic behavior of the
system.

*3.2. Interactions between surface and subsurface in Rohrschollen Island*

Once calibration and validation were completed, the ability to capture the interactions

between surface and subsurface was checked by comparing the modeled exfiltration patterns
simulated on January 22, 2015, with the thermal anomalies identified via airborne TIR
imaging performed the same day (see Section 2). In Fig. 7, the thermal anomalies are
represented as pink spots, and the simulated exfiltration patterns are represented as colored





patches ranging from blue to red as a function of the exfiltration rate. Fig. 7 focuses on the
area of the Island where a vast majority of the thermal anomalies were identified. The
simulated exfiltration patterns usually coincide with the thermal anomalies from the TIR,
even though their spatial extension may be wider than thermal anomalies. This feature can be
the consequence of multiple factors, such as (a) the substantial sedimentary heterogeneity of
the streambed not sufficiently represented in the model, (b) a spatial resolution of the
computation mesh not fine enough to capture the very small-scale surface-subsurface
interactions, and (c) the measurement uncertainty plaguing the TIR analysis. Keeping these
approximations in mind, the hydrologic model correctly locates the surface-subsurface
interactions in the Island and provides flux values that are not accessible via TIR survey.

Fig. 8 and Fig. 9 picture the transient interactions between surface and subsurface and

tell us why the banana-shaped exfiltration zone reported in Fig. 7 is close to the junction of
the new artificial channel and the BGW. Fig. 8 displays maps of the groundwater head, the
surface water thickness, and the exfiltration rates over the whole Island at three different times
of the calibration period that are  t = 50 h (i.e., after the first injection peak);  t = 59 h (i.e., at
the second injection peak); and t = 1072 h (i.e., the date of the airborne TIR flight). As
evidenced by the snapshots of groundwater head and surface water thickness, the water
injected upstream to the island, flowing into the BGW, its dead-ends, and the associated
floodplain, rapidly infiltrates, producing an important increase in groundwater levels
alongside the new artificial channel (and also the BGW), which had been excavated but was
still not clogged with fine sediments. When the maximum injection rate is reached (t = 59 h),
surface ponding occurs on a significant portion of the Island and the groundwater mounding
invades all the upstream part of the BGW. Note that the exfiltration rates (Fig. 8, right) are
localized in small topographic depressions during the injection period, and the banana-shaped
exfiltration pattern (Fig. 7) is still inactive. The latter pattern only appears during the

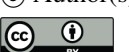



recession period (t = 1072 h) when the strong injection rates have stopped. It appears
alongside the BGW in the vicinity of the area where the groundwater level previously
increased the most. Fig. 9 represents cross-sections along locations *a* and *b* in Fig. 7 for t = 59
h and t = 1072 h, and reports on the subsurface water head, the surface water elevation (set to
the topography elevation when surface water thickness is zero), and infiltration-exfiltration
rates. It shows that (a) the topography mainly controls the banana-shaped infiltration-
exfiltration zone (depressions in Fig. 9), and (b) the temporal dynamics and amplitude of
exfiltration are the combined effect of surface water rapidly flowing toward the system outlet
(i.e., surface water thickness diminishes), and a slow recession of the groundwater heads after
the main peaks of injected flow rates have vanished.

Fig. 10 reports on the evolution over time of the total infiltration and exfiltration

fluxes calculated over the whole surface area of the Island during the two peaks calibration
period. While the injection rate is kept at 2 m$^3$ s$^{-1}$, both infiltration and exfiltration fluxes are
stable with much more infiltration than exfiltration. When the injected flow rate increases, the
infiltrated flux follows a slightly delayed evolution over time, which is very similar to the
injection hydrograph (with a two peaks shape, see Fig. 3). Meanwhile, as the hydraulic
gradient between surface and subsurface changes at some locations, the exfiltration decreases
in areas that turn from an exfiltration to an infiltration regime due to excess of surface water
associated with injection peaks. Once the injection of water into the new artificial channel
stops, the infiltration flux sharply decreases while the exfiltration flux increases. An
exfiltration peak can be observed just at the end of the recession period. It is noteworthy that
during the recession period, the exfiltration flux is almost constant over time and kept at a
value twice that observed before injection (Fig. 10). In the end, forced water injections at the
new channel inlet foster water exfiltration from the subsurface that maintains ponds and



wetlands on the surface over long periods (say, approximately 15 days for each injection, as
simulated by the model but not reported in Fig. 10).

*3.3. Efficiency of the restoration actions*
One of the issues targeted in this study is to assess the efficiency of hydrological
restoration projects. The previous results indicate that water injections in the new channel
enhance the interactions between surface and subsurface compartments of the Island, noting
that the new channel was excavated in highly conductive sedimentary formations. It may be
interesting to check via a modeling approach what causes differences between the current
restored circumstances and a pre-restoration situation. As the pre-restored island is not well
documented in terms of hydraulic data, we considered a scenario where the pre-restored
island is similar to the current situation (including, e.g., geometry and boundary conditions)
with the exception that the newly excavated channel connecting Rohrschollen Island's BGW
and the Rhine River is absent. Therefore, no forced injection may occur at the southern
boundary of the pre-restored island. The hydrological behavior of the pre-restored situation
has been simulated and compared with an actual case where the injection rate in the new
channel is at the usual year-round configuration of 2 $m^3s^{-1}$.
Fig. 11 displays snapshots of exfiltration rates in a subarea of the Island for the pre-
restored and the restored scenarios. Even with an injected flow rate of 2 $m^3s^{-1}$, both the
exfiltration surfaces and exfiltration rates are much higher in the restored situation. In other
words, the base flow regime of the restored situation is sufficient to positively impact the
interactions between surface and subsurface compartments of the Island. When forced
injections enhance the development of wetlands and maintain high rates of exfiltration over
long periods, from the mere hydrological standpoint, restoration works are successful.



*3.4. Suggestions for management practices*

The injection scenarios tested in the hydrological model with maximum peaks reaching 80 $m^3s^{-1}$ are designed as a routine inlet for feeding Rohrschollen Island with water, but some other inlet procedures can also be considered to improve the functioning of the Island. We analyzed with the hydrological model how these routine injections could be designed to maximize either the spatial extension of exfiltration areas maintaining wetlands in surface or the time over which exfiltration occurs. Two hypothetical injections superimposed to a base flow of 2 $m^3s^{-1}$ in the new channel were proposed, the first one being of short duration (24 h) with an injection rate of 15 $m^3s^{-1}$, the second one being of longer duration (120 h) but with a weaker injection rate of 5 $m^3s^{-1}$ (see Fig.12, up). As the total injected water volume differs between both scenarios (the weaker injection flushes almost twice the volume of the stronger injection), it can also be determined which of the two configurations—high rate/small volume or small rate/high volume—maximizes extension and/or duration of exfiltration.

Fig. 12 (down) plots the excess or lack of exfiltration surface areas during injections compared with surface areas sustained by base flow (2$m^3s^{-1}$) in the new channel. The evolution over time of these excess exfiltration areas (or lack thereof) occurs for both injection scenarios with a lack of exfiltration areas occurring during the injection periods when infiltration from the surface dominates. After the injection peak is completed, the recession period—starting at t = 52 h for the high injection rate and t = 162 h for the small injection rate (Fig. 12)—always shows an excess of exfiltration areas. The interesting point is that the high injection rate delivers a smaller volume of water in the system but maintains increased areas of exfiltration over extensive periods. For its part, the small injection rate has no effect beyond t = 250 h with a system coming back to its initial state with 2$m^3s^{-1}$ of routine injection at the inlet. Finally, injecting less volume but with high injection rates over short



periods is better suited to maintaining exfiltration over long periods as the process feeding
wetlands on the Island (Fig. 12). It is also likely (though not studied in this work) that intense
injections favor the unclogging of the BGW, which are the primary surface water routes
contributing to water renewal on the Island.

**4. Conclusions**

Restoration projects to counterbalance the undesired effects of anthropogenic actions

on the hydrological, geomorphological, and ecological status of riverine ecosystems have
recently spread worldwide. As the interactions between surface and subsurface compartments
of the hydrosystem have a strong impact on hydrological, biogeochemical, and ecological
processes, it makes sense to rely upon integrated hydrological modeling when addressing the
question of restoration efficiency. When feasible (i.e., with tractable problems and models),
hydrological modeling with high resolution in time and space can accurately delineate
infiltration-exfiltration areas and their evolution over time as key factors for maintaining
active surface river networks

Relying upon simplified models, not in their physics but rather on their dimensionality

(as done in the present study), renders many problems tractable and calculable. This is the
case with Rohrschollen Island, which shows smooth variations of topography that do not help
to locate ground water outcrops. This comment also extends to the very transient hydraulic
behaviors requiring refined time steps to accurately capture temporal evolutions of the
system.

If the focus is placed on infiltration-exfiltration patterns as a reliable indicator of the

effects of restoration in riverine systems, any spatially distributed modeling exercise needs
conditioning regarding both model inputs and outputs. Concerning the conditioning (or
control) of model outputs associated with the delineation of exfiltration areas, the recent

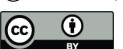



technique of airborne, low altitude, and high-resolution thermal infrared imaging is very
promising. The technique is not free of measurement errors and artifacts, but it has been
shown reliable enough to highlight interactions between surface and subsurface compartments
of the hydrosystem that coincide with simulations. Further investigations should duplicate
thermal imaging over time with the aim of grasping the transient behavior of surface-
subsurface interactions and discussing the best versus the worst environmental conditions
where imaging is applicable.

Rohrschollen Island (and many other fluvial hydrosystems) is very specific regarding

surface-subsurface interactions, meaning that water heads in the aquifer are often close to
surface water levels. This means that slight variations in both compartments may invert the
direction of exchanged fluxes between compartments. In that case, injecting significant
volumes of water in a system to store them over large periods may be counterproductive, even
though these volumes may contribute to flooding over large areas. Large volumes are diverted
into the rapidly flowing surface water and exit the system. Intense injections of smaller
volumes over short periods foster intense local infiltration into the subsurface. The subsequent
water mounding in the aquifer then results in long-term storage and smooth release of water
via exfiltration. This behavior, hardly foreseeable, was that simulated for Rohrschollen Island
and could also apply to many other configurations of fluvial corridors. These results show that
management rules for a restored system may be developed from modeling exercises handling
various forcing scenarios applied to the system. If it is accepted that exfiltration (sustaining
ponding and wetlands) is a valuable indicator of riverine restoration, additional works should
envision various settings to improve this process. For example, it is not clear if several
smaller inlets could replace a single inlet in the system for higher efficiency. Is water
extraction from the surface and reinjection in the subsurface a valuable process that can





generate slow exfiltration over broad areas? Physically-based integrated modeling of
hydrosystems might propose some answers.

**Acknowledgements**
The monitoring of the Rohrschollen Island was funded by the European Community (LIFE08
NAT/F/00471), the City of Strasbourg, the University of Strasbourg (IDEX-CNRS 2014
MODELROH project), the French National Center for Scientific Research (CNRS), the
ZAEU (Zone Atelier Environnementale Urbaine - LTER), the Water Rhin-Meuse Agency, the
DREAL Alsace, the "Région Alsace," the "Département du Bas-Rhin," and the company
"Électricité de France." The authors are also indebted to Pascal Finaud-Guyot for his
contribution in the preprocessing of hydrological datasets.



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



**Figure captions**

Fig. 1. (a) location of the studied area (France), (b) aerial view of Rohrschollen Island, and (c) network of hydrologic response measurements (mainly hydraulic heads and water fluxes).

Fig. 2. Digital elevation model of Rohrschollen Island (left) and location of the main gravel bars reconstructed from the geo-historical and sedimentological studies (right). The black and white lines correspond to transects of hydrologic measurements (see Figure 1).

Fig. 3. Evolution over time of flow rates injected in the new artificial channel feeding Rohrschollen Island during the period selected for calibrating the integrated hydrological model (up), and the period chosen as a validation (forecasting) exercise (down).

Fig.4. Calibrated fields of saturated hydraulic conductivity in the subsurface compartment (left) and exchange coefficient between surface and subsurface compartments (right).

Fig. 5. Comparison between simulated and measured hydraulic heads in the subsurface during the calibration period. Left: evolution over time at the two transects, that is, the worst (up) and best (down) transects regarding RMSE. Right: Local in space and time values of simulated hydraulic heads as a function of observed ones. RMSE = root of mean square error, and KGE = Kling-Gupta efficiency.

Fig. 6. Comparison between simulated and measured hydraulic heads in the subsurface during the validation period. Left: evolution over time at the two transects, that is, the worst (up) and best (down) transects regarding RMSE. Right: local in space and time values of simulated hydraulic heads as a function of observed ones. RMSE = root of mean square error, and KGE = Kling-Gupta efficiency.

Fig 7. Comparison between simulated exfiltration patterns and thermal anomalies identified via thermal infrared imaging close to the junction between the new channel (southeast corner) and the BGW (Bauerngrundwasser; center of Fig.). Red transects a and b are locations where surface water and groundwater head are followed to exemplify surface-subsurface interactions in Fig 9.

Fig. 8. Groundwater head, surface water thickness, and exfiltration rate over the whole of Rohrschollen Island for three different periods (in hours after the beginning of injection) of the calibration period. Notably, the last period is also the date of the airborne thermal infrared imaging.

Fig. 9. Evolution of surface water elevation (blue), groundwater head (red), and exchange fluxes (arrows) along transects a and b (located in Fig. 7) at two periods (hours after the beginning of injection) of the calibration period. A thick grey line represents the topographic profile. The grey scale indicates values of the saturated hydraulic conductivity at the interface between surface and subsurface.

Fig. 10. Evolution of the infiltration and exfiltration volumetric fluxes during the first steps of the calibration period (where evolutions are essential).

Fig. 11. Patterns of exfiltration for the pre-restored and the restored situations. The focus is on the most active zone of Rohrschollen Island regarding surface-subsurface interactions.



Fig. 12. Up: injection rates of two scenarios seeking optimal exfiltration surface areas and
durations at Rohrschollen Island. Down: Evolution over time of excess or lack of exfiltration
surface area compared with exfiltration surface produced by a routine injection rate of 2 $m^3\,s^{-1}$
at the inlet of the system.





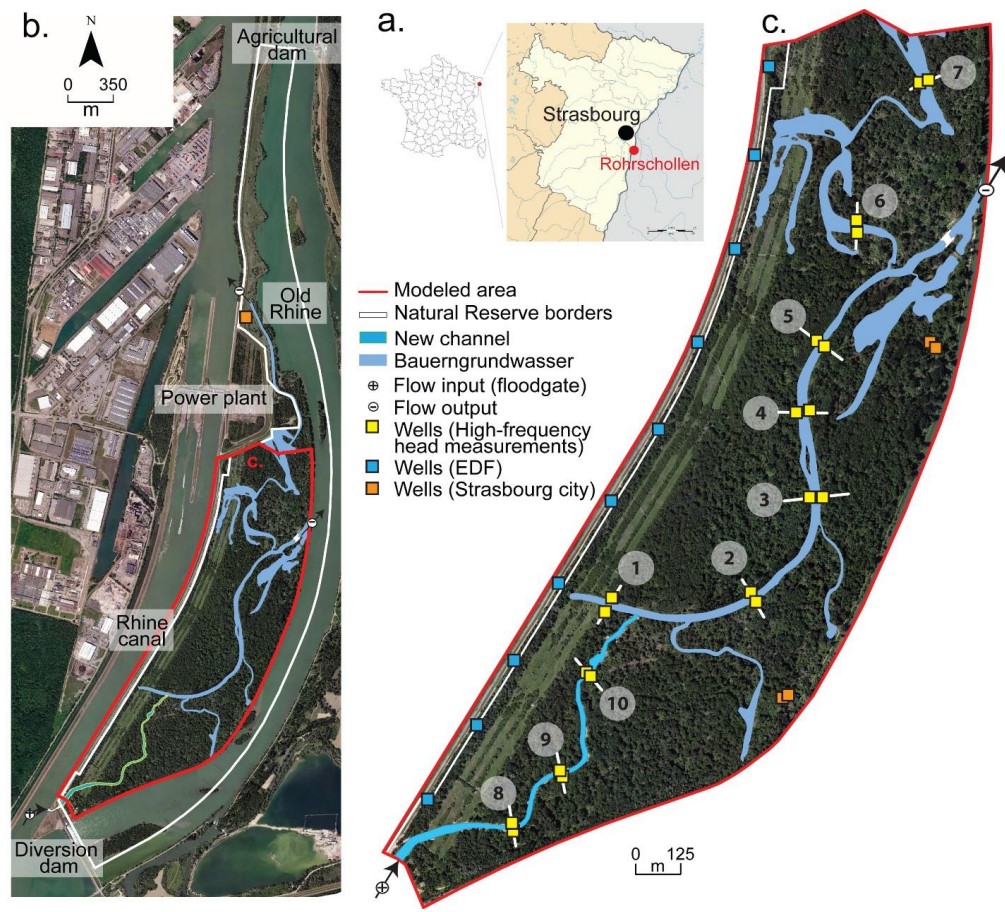



**Fig 1.** (a) location of the studied area (France), (b) aerial view of Rohrschollen Island, and (c) network of hydrologic response measurements (mainly hydraulic heads and water fluxes).







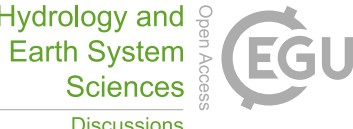



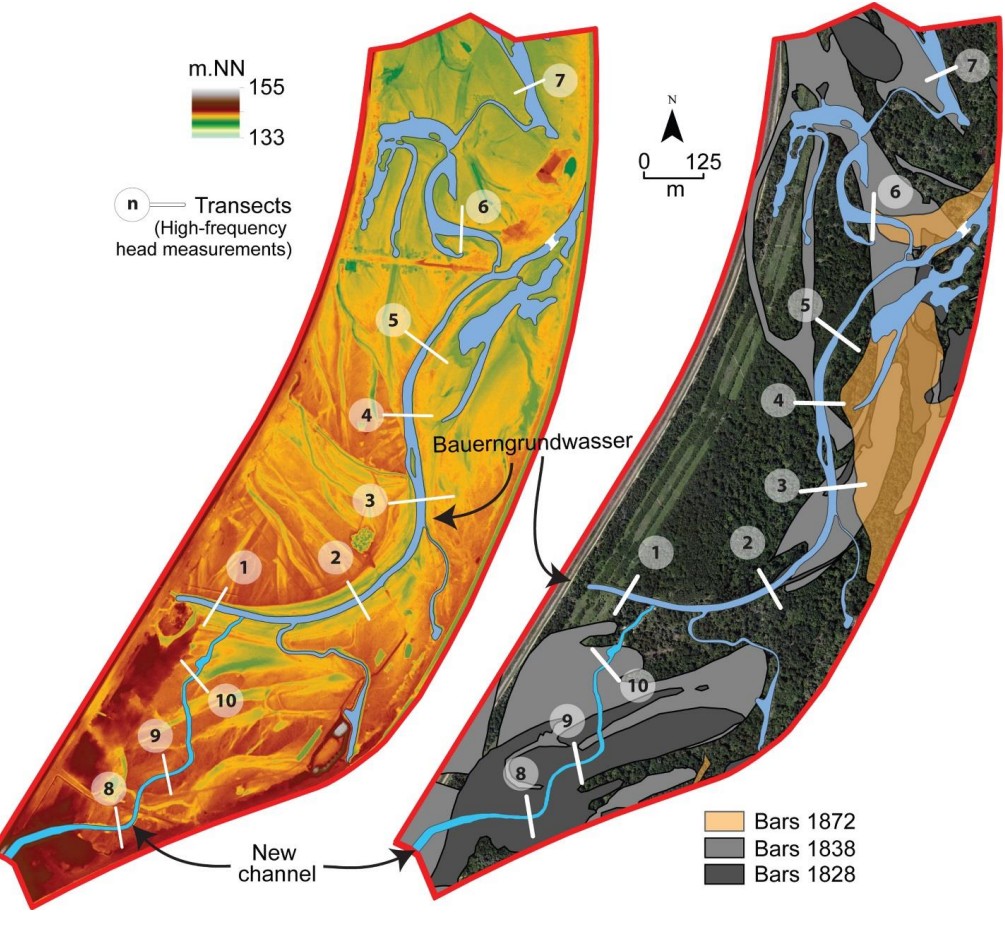



**Fig 2.** Digital elevation model of Rohrschollen Island (left) and location of the main gravel
bars reconstructed from the geo-historical and sedimentological studies (right). The black and
white lines correspond to transects of hydrologic measurements (see Figure 1).










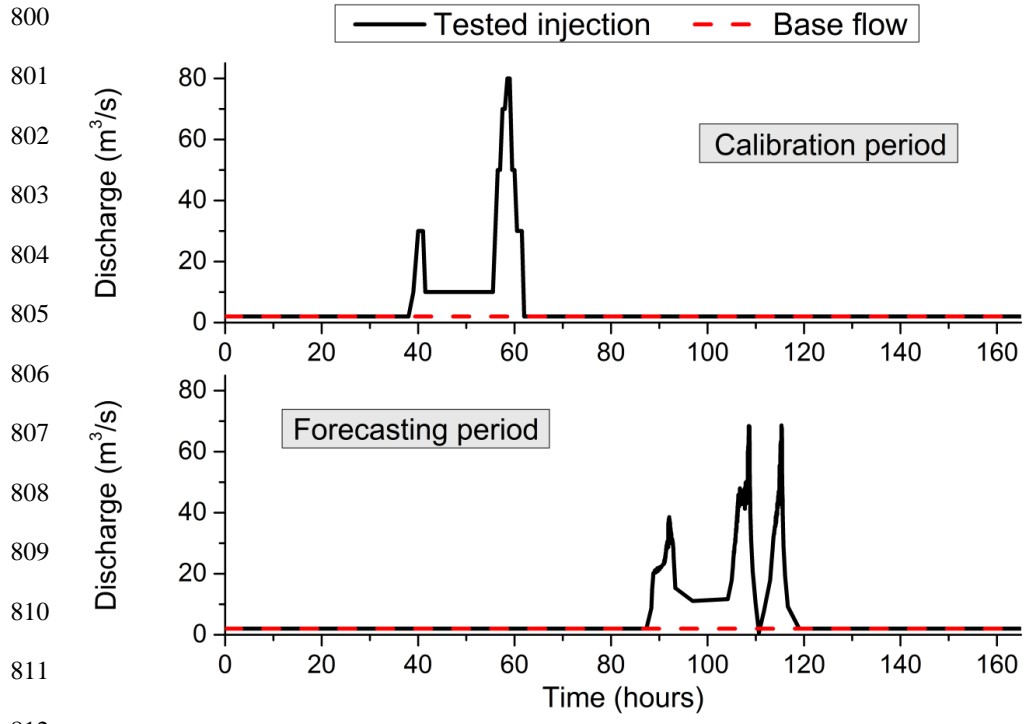

**Fig. 3.** Evolution over time of flow rates injected in the new artificial channel feeding Rohrschollen Island during the period selected for calibrating the integrated hydrological model (up), and the period chosen as a validation (forecasting) exercise (down).





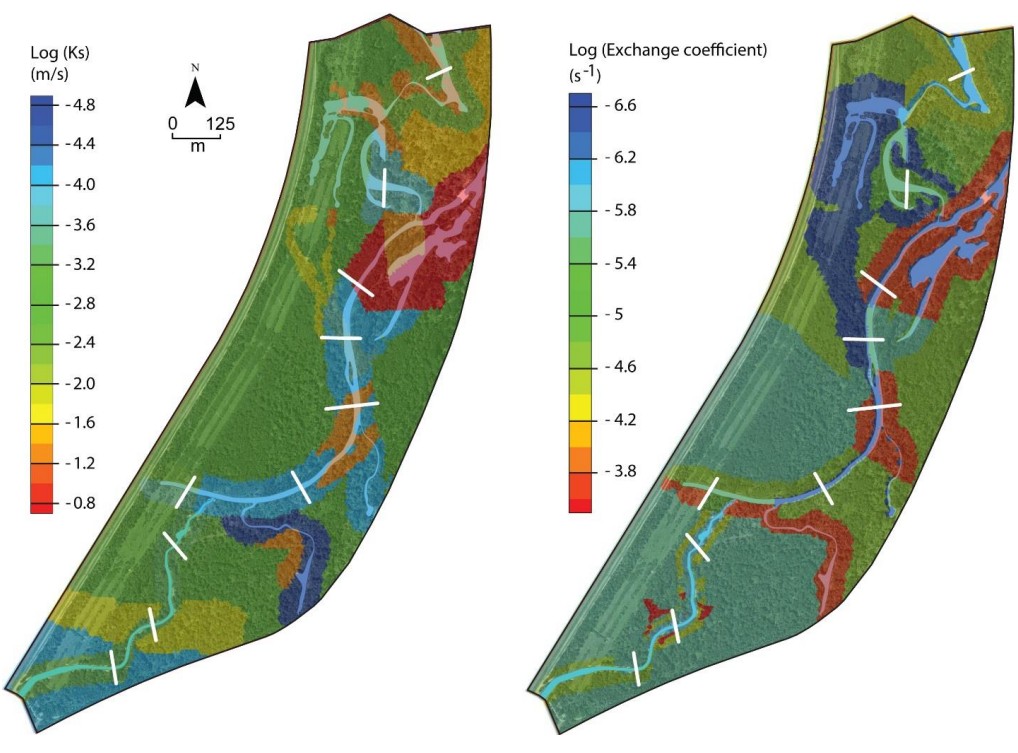



**Fig 4.** Calibrated fields of saturated hydraulic conductivity in the subsurface compartment
(left) and exchange coefficient between surface and subsurface compartments (right).








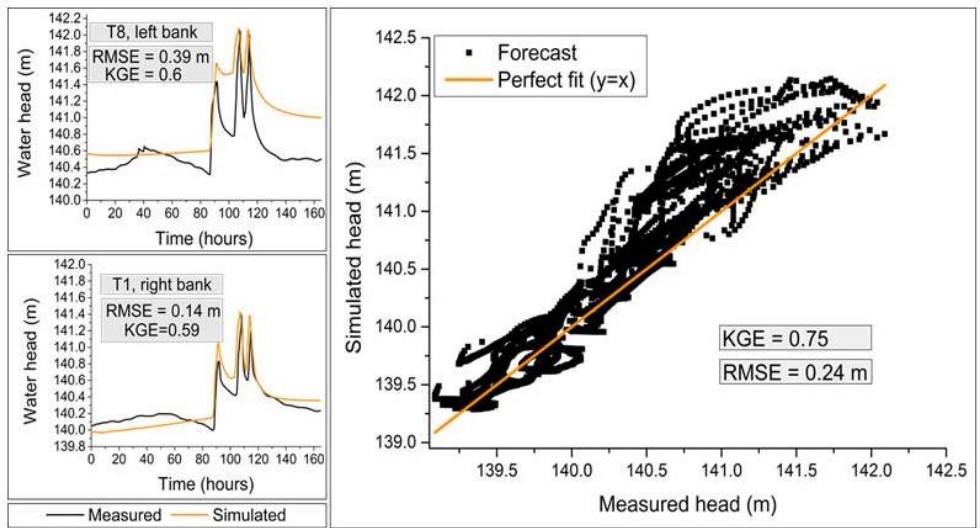

**Fig. 5.** Comparison between simulated and measured hydraulic heads in the subsurface during
the calibration period. Left: evolution over time at the two transects, that is, the worst (up) and
best (down) transects regarding RMSE. Right: Local in space and time values of simulated
hydraulic heads as a function of observed ones. RMSE = root of mean square error, and KGE
= Kling-Gupta efficiency.






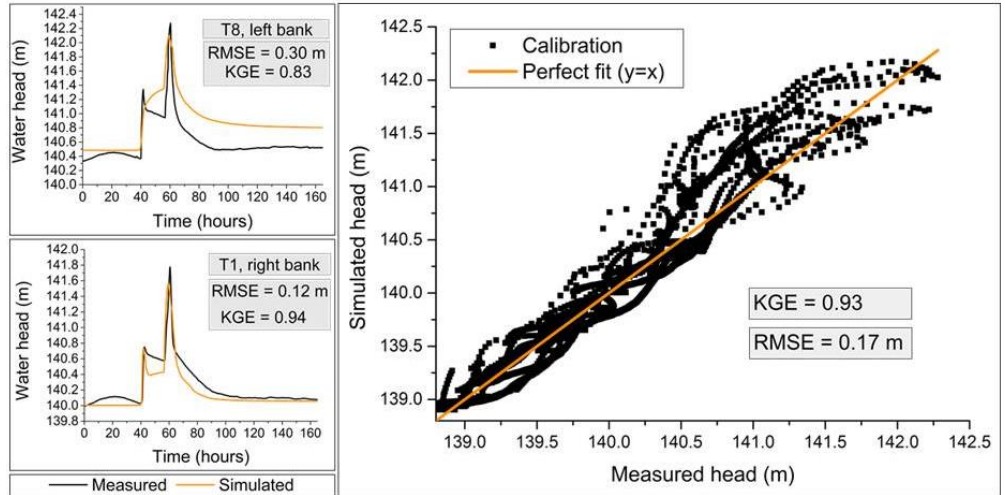



**Fig. 6.** Comparison between simulated and measured hydraulic heads in the
subsurface during the validation period. Left: evolution over time at the two transects, that is,
the worst (up) and best (down) transects regarding RMSE. Right: local in space and time
values of simulated hydraulic heads as a function of observed ones. RMSE = root of mean
square error, and KGE = Kling-Gupta efficiency.






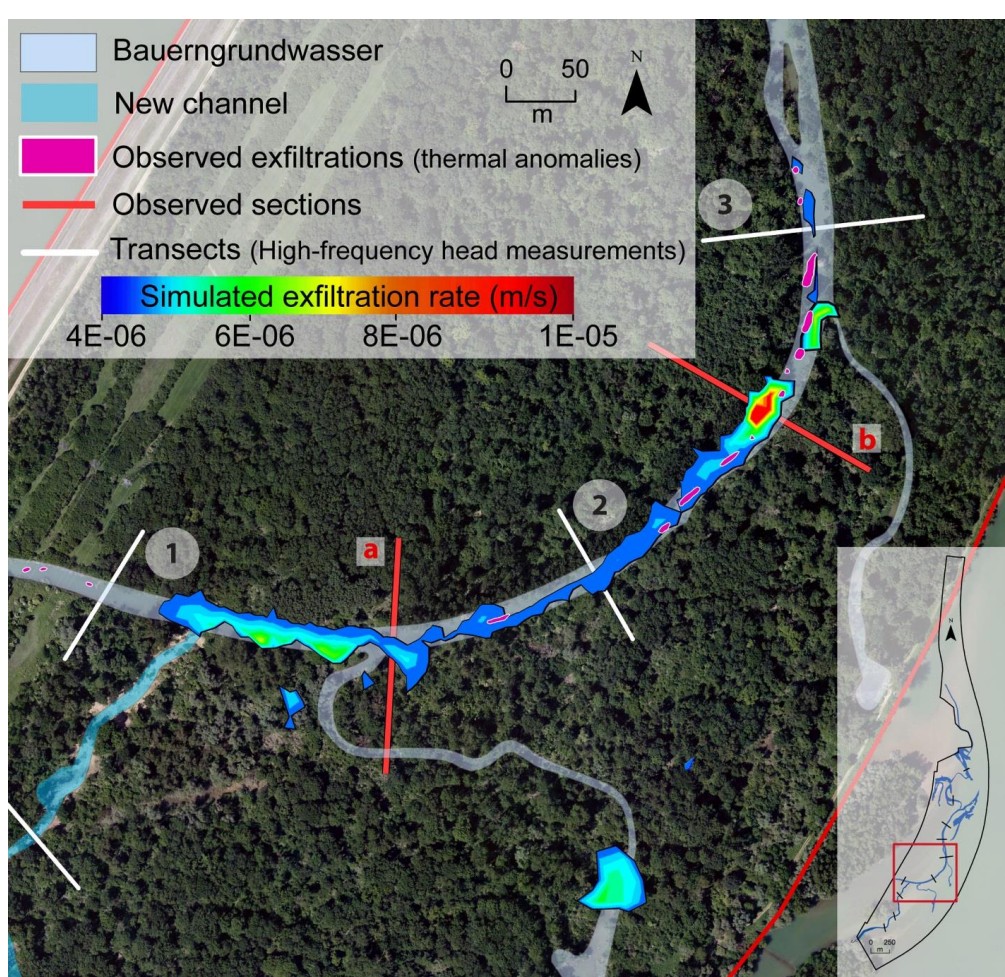

**Fig 7.** Comparison between simulated exfiltration patterns and thermal anomalies identified
via thermal infrared imaging close to the junction between the new channel (southeast corner)
and the BGW (Bauerngrundwasser; center of Fig.). Red transects a and b are locations where
surface water and groundwater head are followed to exemplify surface-subsurface interactions
in Fig 9.







**Fig 8.** Groundwater head, surface water thickness, and exfiltration rate over the whole of
Rohrschollen Island for three different periods (in hours after the beginning of injection) of
the calibration period. Notably, the last period is also the date of the airborne thermal infrared
imaging.





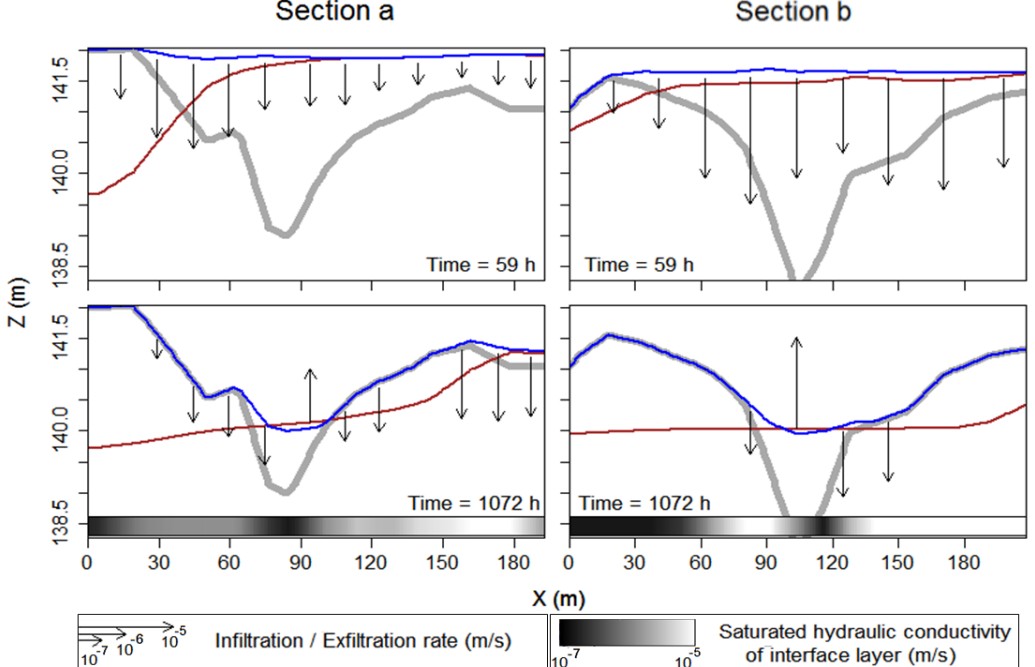

**Fig. 9.** Evolution of surface water elevation (blue), groundwater head (red), and exchange
fluxes (arrows) along transects a and b (located in Fig. 7) at two periods (hours after the
beginning of injection) of the calibration period. A thick grey line represents the topographic
profile. The grey scale indicates values of the saturated hydraulic conductivity at the interface
between surface and subsurface.



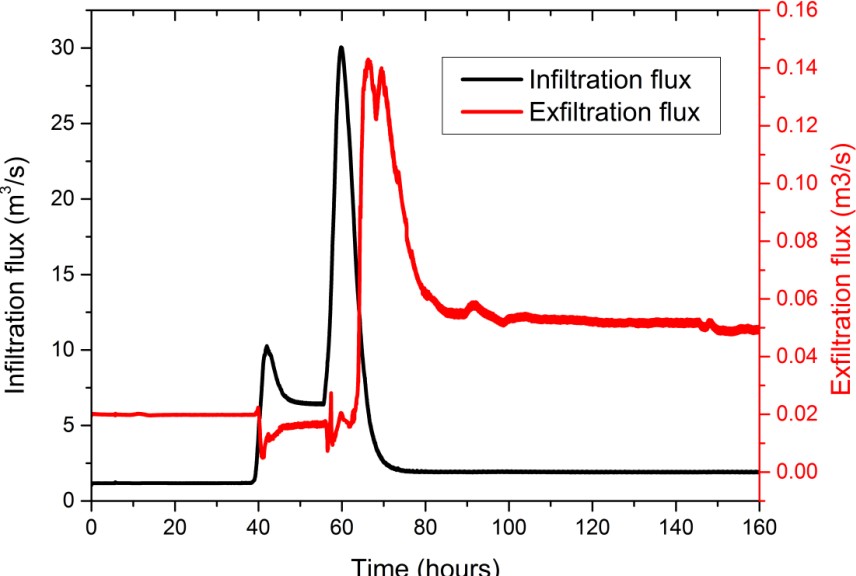


**Fig 10.** Evolution of the infiltration and exfiltration volumetric fluxes during the first steps of
the calibration period (where evolutions are essential).






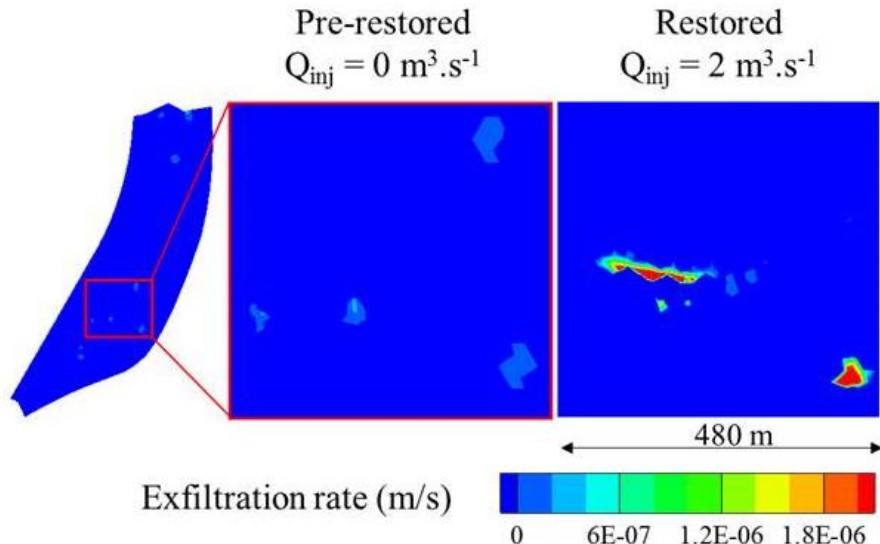


**Fig. 11.** Patterns of exfiltration for the pre-restored and the restored situations. The focus is on
the most active zone of Rohrschollen Island regarding surface-subsurface interactions.








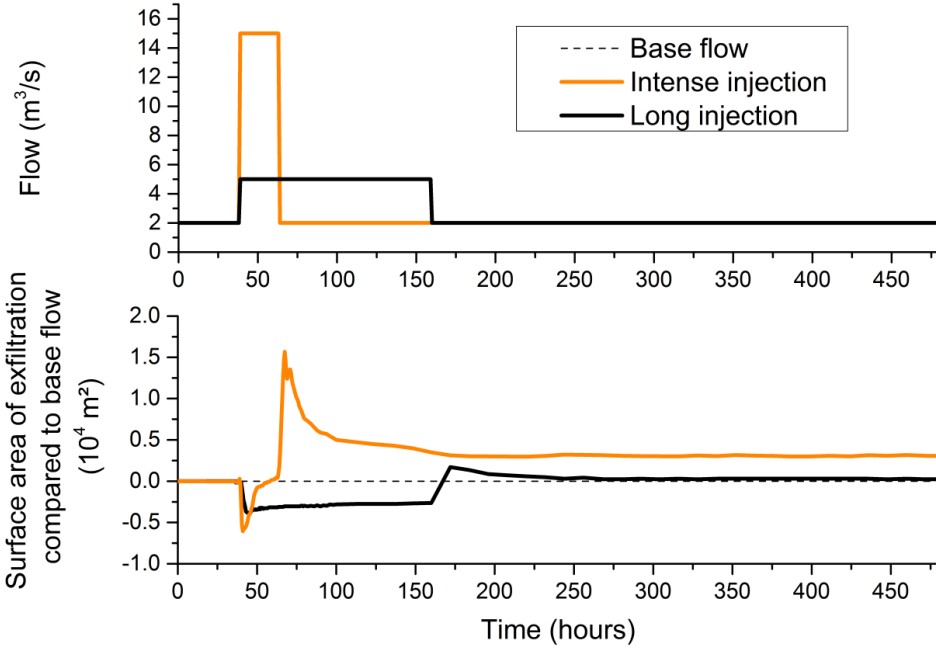

**Fig 12.** Up: injection rates of two scenarios seeking optimal exfiltration surface areas and
durations at Rohrschollen Island. Down: Evolution over time of excess or lack of exfiltration
surface area compared with exfiltration surface produced by a routine injection rate of 2 m$^3$ s$^{-1}$
at the inlet of the system.

