# Peer review of "Assessing the effect of flood restoration on surface-subsurface interactions in Rohrschollen Island (Upper Rhine River – France) using integrated hydrological modeling and thermal infrared imaging"

_Hydrology and Earth System Sciences, 2018_

## Referee Comment (RC1) · Anonymous Referee #1 · 17 Oct 2018

Thank you for giving me the opportunity to review the work by Jeannot and co-authors "Assessing the effect of flood restoration on surface-subsurface interactions in Rohrschollen Island (Upper Rhine River – France) using integrated hydrological modeling and thermal infrared imaging". The authors have set up a surface-subsurface physically-based model to investigate the effects of flood restoration on an island of the Rhine river. After manually calibrating the model on groundwater heads following a flood event (injection), the authors have validated model parameters and their hy-

potheses on groundwater heads of another flood event, and further checked that modeled exfiltration patterns matched observations from airborne thermal infrared imaging. This allowed them to study the mechanisms of groundwater exfiltration in restored conditions, and to compare those results to a simple case of pre-restoration conditions. They have showed that in this case restoration indeed enhanced groundwater exfiltration. They further compared two injection scenarii, high rate/small volume or small rate/high volume, and showed that injecting less water but with high rates over short periods maintained exfiltration over longer periods due to the modeled processes (time scale differences between surface water and groundwater response to floods).

This very well-written manuscript, shows in a short and concise way through this case study how a surface-subsurface hydrological model can be used to investigate complex interactions involving short wavelength and small amplitude topography, fast (overland flow) and slow (groundwater) compartments. The authors did a good job in describing clearly the processes at the origin of the surface-subsurface exchanges directions and amplitudes, backed-up by appropriate figures. Although I have some minor comments which I think should be adressed before publication, I think this work is of high quality and suitable for publication in HESS. I think this is a significan step toward improving tools for bridging the gap between hydrological research and water management stakeholders.

The two main points I want to raise are related to hypotheses which need to be mentioned or discussed: 1) Why setting-up a lower cost numerical model by simplifying the Richards equation if this does not result in using stochastic methods for calibration, or any other specific advantage? Please also give the runtime for those simulations. For instance, having a low cost model could allow sensitivity analysis to help showing which features of the restoration or paleo-geomorphology mostly impact the exchanges. I could also help to give an uncertainty estimates to the soil parameters and to the restoration effects, which could further help stakeholders. 2) I am probably biased, but I think a surface-subsurface hydrological model with no surface boundary condition or

source/sink term has to be justified. While the high rates, volume and resistance coefficients together with rather short periods involved in the present study probably moves evapotranspiration uptakes to a lower order, and one may assume that no rain happened during the studied periods, those points need to be written down and eventually discussed, for instance for future applications where such a model could be applied over longer periods. Also the calibration and validation periods concern different season, likely to be under different evapotranspiration regimes. Also was the vegetation – and ET uptakes- the same before and after restoration? Although I agree that it is likely that ET has minor effect in this study, those points need to be discussed or mentioned. Finally, I also find curious that no mention to the impacts of surface-subsurface exchanges on ecosystem services for the specific case of the Rohrschollen island are discussed in the introduction, which is rather general, while the case study aspect of the paper clearly appears in the manuscript title.

Specific points:

- Introduction: could you provide exemple of hyporheic processes that have specific ecological importance for the Rohrschollen? Because the study site is in the title, I would expect some mentions to it in the introduction, which is overall a little long and vague... maybe cutting off some repetitions and adding up a brief section on the specific targets of the restorations on the ecosystems of the Rohrschollen would encourage the reader? This is just a suggestion to improve the quality of the paper. - L162: "degraded the hydrological, geomorphological, and ecological functioning of the hydrosystem." This line and the following sentences would benefit from indicating a proper reference. Consider using 'impacted' instead of degraded if no detailed description (or reference) of those functioning can be given. - Fig. 3: What is the baseflow in this case ? How is it obtained ? It is not discussed in the main text. - Fig 2 and 4 are not really the same... how have you decided to change the spatial structures ? - Fig. 5. KGE: I understand that you show both RMSE and KGE, because KGE does not change much when the simulation clearly matches less to the observation. This is

probably due to compensations in the KGE terms and can be discussed in the main text by giving the three KGE terms (variance and bias ratio, and correlation coefficient). - Fig. 5 and 6.: What are the boundary conditions time series on West and East sides ? Would it be relevant to show it? How much of the water comes from the side and how much from the flood? It could help to add up West and East bank boundary conditions and injection time series in this figure, for instance by lowering the size of the scatterplot. - L 297: "Results from particle size analysis also helped to predefine variation ranges of crucial parameters, such as the hydraulic conductivity and retention curve parameters of the sediments and the exchange coefficient between surface and subsurface.": This is key, have you any validation data of the calibrated values? Particularly over the different patches? Did you use pedotransfers functions? Which ones?

---

## Referee Comment (RC2) · Anonymous Referee #2 · 24 Oct 2018

Review of the manuscript hess-2018-439 "Assessing the effect of flood restoration on surface-subsurface interactions in Rohrschollen Island (Upper Rhine River – France) using integrated hydrological modelling and thermal infrared imaging" for Hydrology and Earth System Sciences

Jeannot et al. in the manuscript "Assessing the effect of flood restoration on surface-subsurface interactions in Rohrschollen Island (Upper Rhine River – France) using

integrated hydrological modelling and thermal infrared imaging" evaluate the efficiency of restoration actions adopted in Rohrschollen Island, in the Upper Rhine River specifically in terms of surface-subsurface flow exchange in the hyporheic zone. The surface-subsurface interaction is quantified with a fully-distributed hydrological model, the Normally Integrated Model (NIM). The exfiltration areas are of great interest for this study, as they represent the opportunity for hyporheic exchange enhancement. Using an innovative approach, the output of the validated model in terms of exfiltration is compared with the information derived from thermal infrared imaging.

General comments

I appreciated reading this manuscript, I found it very instructive and I think it is a valuable contribution to HESS. Authors address a key issue for ecological restoration and suggest an innovative framework to evaluate the impact of anthropogenic activities. I have some suggestions I would like to see addressed in a revised manuscript. My first observation concerns the model setup. I might have misunderstood, but the surface-subsurface flow interaction model seems to be not considering some input/output terms related to the surface processes, e.g., the precipitation and evapotranspiration. If they are somehow included in the model parameters this should be pointed out in the manuscript and, if not, this hypothesis should be clearly stated and justified. Another point related to the model setup concerns the time step and the time horizon of the simulation. These aspects are not discussed at all in the presentation of the model setup. Being this a study of the effects of some management policies, the reader expects a long-term problem setup. Moreover, the time step and the time horizon might influence the results and this should be discussed in the manuscript. In other words, the short time horizon implies calibrating the model on one event only, opening a debate about the robustness of the calibration, although the performance in validation might be convincing.

My second observation concerns the sensitivity of the results of the hydrological model to the calibrated parameters, such as the hydraulic conductivity. The authors claim

that the exfiltration areas are the result of rapid infiltration which produces an important increase in groundwater level. However, the exfiltration areas do not always coincide with the temperature anomalies observed in the infrared imaging. The authors point out possible causes of the mismatch but do not talk about the sensitivity of the model to the calibrated parameters. One could claim that the observed results might be due to an overestimation of the hydraulic conductivity in certain specific areas of the catchment. The discussion of this issue is required in the revised version of the manuscript.

Specific comments

L84-93: It would be interesting to have some examples of how increasing exchanges in the hyporheic zone contributes to the restoration projects.

L109: Wrong citation: "Fattichi" change into Fatichi.

L115: change computer into computational.

L124-126: Refer citations to the specific effect which is taken into account (water table dynamics, flood frequency, ecosystem services..).

L138-142: Better to express research objectives, possibly related to the discussion paragraphs: (1) model performance, (2) comparison between model results and TIR images and (3) comparison of different management options in terms of input quantity (and frequency, to add).

L296-299: The results from field experiments helped predefining the ranges of variations of some parameters of the model, but still some uncertainty exist on their calibration. It is worth commenting on the uncertainty coming from the calibration of the parameters of the model here and/or in the discussion section. Moreover, some parameters that I assume were calibrated (e.g., the Manning coefficients) are not mentioned in Section 2.2.2. Maybe a table with the starting range of variation of all the calibrated parameters and the calibrated value could be useful.

L277-305: Here the simulation time step and time horizon should be discussed, because the reader does not know which is the reference time scale, also because restoration processes are usually associated with long time scales.

L315: "After a first simulation employing the initial parametrization (defined in Section 2.2.2)" see comments below concerning the explanation of the model parametrization. It should be more exhaustive.

L320-322: "Only the hydraulic conductivity and the exchange coefficient between surface and subsurface were slightly adjusted while trying to preserve the initial spatial zonation" Not clear statement. Which is the initial spatial zonation? How was it defined? Does it mean hydraulic conductivity and the exchange coefficient are the only manually-calibrated parameters? What about the other parameters of the model? How were they fixed?

L341-360: The discussion is consistent, but Figure 5 and Figure 6 are switched, so Figure 5 refers to the validation and Figure 6 to the calibration. This Section lacks of comments on the impact of the simulation horizon. The calibration on a single event does impact the results of calibration and validation. Please, comment on this.

L354 and L357: It would be interesting to see the values of the three components of the KGE and their variation from calibration to validation.

L371-377: Among this factors also the sensitivity to the model parameters should be pointed out. Somebody could claim that the observed dynamic might be due to an overestimation of the hydraulic conductivity implying higher infiltration and consequently exfiltration on a much larger area than the one where thermal anomalies are observed. Moreover, it is difficult to quantify the uncertainty related to the airborne TIR images, which were collected in a single survey. Maybe some comments on this uncertainty might help the reader to evaluate the results robustness.

L406-408: It could be helpful adding in Fig.10 also the injected flow reported in Fig.3.

L424: "..noting that the new channel was excavated in highly conductive sedimentary

formations" this information comes from field experiments or from the parameters of the calibrated model?

L464: "..but maintains increased areas of exfiltration over extensive periods" the definition "extensive periods" would have a more precise meaning if the problem in terms of time horizons was discussed while defining the boundaries of the problem. The observation is suitable also for "long periods" at L467. Section 3.4 is very interesting because it tests two different mitigation strategies in terms of input rate and volume, but under the title "Suggestions for management practices" some more information is expected, for example in terms of exchange frequency required over one year in order to observe ecological enhancement. Adding some information in this direction completes also the conclusions of Section 3.3, where the authors state "When forced injections enhance the development of wetlands and maintain high rates of exfiltration over long periods, from the mere hydrological stand point, restoration works are successful", but how often does it happen? How often should it happen in order to enhance the ecological status of the environment?

Figure 5: switch with Figure 6.

Figure 10: Add the pattern of the inflow. Right y-axes change m3/s into m3/s. If possible, it would be nice to have enhanced image quality of Figure 5, 6, 8, 9 and 11.

---

## Author Comment (AC1) · 9 Nov 2018

"Assessing the effect of flood restoration on surface-subsurface interactions in Rohrschollen Island 'Upper Rhine River – France) using integrated hydrological modeling and thermal infrared imaging", by B. Jeannot et al.

Reply to comments raised by Rev #1
The discussion reported hereafter remind us with the comments of the Reviewer typed in straight font as our answers appear below in italic. In the reviewing process of HESS, a reply should be sent before being allowed to propose a revised manuscript. This is why some of our answers might appear as declarations of intend to which we would then try to stick in writing the revised version of the manuscript.
* * *
Thank you for giving me the opportunity to review the work by Jeannot and coauthors "Assessing the effect of flood restoration on surface-subsurface interactions in Rohrschollen Island (Upper Rhine River – France) using integrated hydrological modeling and thermal infrared imaging". The authors have set up a surface-subsurface physically-based model to investigate the effects of flood restoration on an island of the Rhine river. After manually calibrating the model on groundwater heads following a flood event (injection), the authors have validated model parameters and their hypotheses on groundwater heads of another flood event, and further checked that modeled exfiltration patterns matched observations from airborne thermal infrared imaging. This allowed them to study the mechanisms of groundwater exfiltration in restored conditions, and to compare those results to a simple case of pre-restoration conditions. They have showed that in this case restoration indeed enhanced groundwater exfiltration. They further compared two injection scenarii, high rate/small volume or small rate/high volume, and showed that injecting less water but with high rates over short periods maintained exfiltration over longer periods due to the modeled processes (time scale differences between surface water and groundwater response to floods).

- *We are grateful to the Reviewer for his (her) sharp and synthetic view on the material constituting this hydrological study. It is right that we mainly focused the modeling task on mimicking the hydrological behavior of the system over short periods of time but associated with very transient flow conditions. This is the main added value of the study, as very transient features are still challenging to model in the various compartments of the system due to contrasted characteristic times between flow processes and the spatial resolution needed to clearly grasp surface-subsurface flow interactions. This motivated the use of an integrated hydrological model in a form reducing the dimensionality of the subsurface compartment with the idea of rendering tractable simulation highly-resolved in time and space. The main drawback is that, in the context of restoration, which can be a long-term objective, we do not completely evaluate the benefits of restoration works, our investigation being limited to grasp how behave water bodies after forced flood periods.*

This very well-written manuscript, shows in a short and concise way through this case study how a surface-subsurface hydrological model can be used to investigate complex interactions involving short wavelength and small amplitude topography, fast (overland flow) and slow (groundwater) compartments. The authors did a good job in describing clearly the processes at the origin of the surface-subsurface exchanges directions and amplitudes, backed-up by appropriate figures. Although I have some minor comments which I think should be addressed before publication, I think this work is of high quality and suitable for publication in HESS. I think this is a significant step toward improving tools for bridging the gap between hydrological research and water management stakeholders.

- *As told just above, our main goal was to address the feasibility of simulating very transient features with highly-resolved models. We thank the Reviewer for his (her) very positive appraisal on the way we handled this task.*

The two main points I want to raise are related to hypotheses which need to be mentioned or discussed: 1) Why setting-up a lower cost numerical model by simplifying the Richards equation if this does not result in using stochastic methods for calibration, or any other specific advantage? Please also give the runtime for those simulations. For instance, having a low cost model could allow sensitivity analysis to help showing which features of the restoration or paleo-geomorphology mostly impact the exchanges. It could also help to give an uncertainty estimates to the soil parameters and to the restoration effects, which could further help stakeholders.

- *We fully agree that, among the various interests associated with simplified (tractable) models, the "Monte Carlo philosophy" duplicating simulations for various purposes such as sensitivity analysis, inverse problems, tests on hypotheses, etc., is very useful. We did not start our study with this option in mind, simply because we ignored how a simplified model could render a valuable hydrological simulation of the system. Exploratory calculations with a fully-dimensioned model showed that modeling the system was cumbersome, because of: 1- the flat topography needing for spatially refined grids to delineate the flooding of wetlands and ponds, and 2- the very contrasted times of response between the surface and the subsurface after forced injections. One may therefore consider that employing our integrated model over Rohrschollen Island is a preliminary test, before further investigations. We envision to couple inversion procedures to the integrated model with the aim of providing equiprobable configurations of hydrological systems that all match up observation data. The revised manuscript could better justify our choice, probably in the Section presenting the "Hydrological modeling strategy" Mean runtimes of simulations on a standard computer (approximately 5 hours to simulate the first 7 days and 24 h to simulate the whole 45 days) could also be given knowing that independent simulations, as performed in "Monte Carlo" approaches, easily benefit from distributing the calculations tasks over the multiple cores of modern processors.*

2) I am probably biased, but I think a surface-subsurface hydrological model with no surface boundary condition or source/sink term has to be justified. While the high rates, volume and resistance coefficients together with rather short periods involved in the present study probably moves evapotranspiration uptakes to a lower order, and one may assume that no rain happened during the studied periods, those points need to be written down and eventually discussed, for instance for future applications where such a model could be applied over longer periods. Also the calibration and validation periods concern different season, likely to be under different evapotranspiration regimes. Also was the vegetation– and ET uptakes- the same before and after restoration? Although I agree that it is likely that ET has minor effect in this study, those points need to be discussed or mentioned. Finally, I also find curious that no mention to the impacts of surface-subsurface exchanges on ecosystem services for the specific case of the Rohrschollen island are discussed in the introduction, which is rather general, while the case study aspect of the paper clearly appears in the manuscript title.

- *Rev#1 feeds our response in his (her) interesting question! As already mentioned the study is focused on mimicking short-term responses associated with flash floods subsequent to forced flow conditions. The volume of water injected in the system and the varying boundary conditions at the banks of a riverine island associated with dam storages and releases in the river are the main features controlling the evolution of*

*water bodies in the Island. Even under a continuous routine base flow injection of 2 $m^3 s^{-1}$ through the artificial new channel, the volume of water brought by the Rhine River to the Island is approximately $6.3 \times 10^7$ m3 in a year, which would correspond to an equivalent infiltration of 15 m of water in a year over the whole Island. Most of the base flow injection infiltrates via the new channel and the BGW (almost no flow exits the BGW to the North), but even with 20% of infiltration of the base flow injection, this would still correspond to 3 m of rainfall infiltration. This rapid calculations (not reported in the manuscript, we agree) led us to consider that rainfall infiltration and ET were negligible within the modeled short periods of intense flooding. Regarding the eventual benefits brought by floods to ecosystem services, we note that the title of the manuscript only mentions the effects on surface-subsurface flow interactions. That being said, we agree that the Introduction could let room for a few sentences regarding ecosystem services. The revised manuscript could be amended accordingly.*

Specific points:
- Introduction: could you provide example of hyporheic processes that have specific ecological importance for the Rohrschollen? Because the study site is in the title, I would expect some mentions to it in the introduction, which is overall a little long and vague… maybe cutting off some repetitions and adding up a brief section on the specific targets of the restorations on the ecosystems of the Rohrschollen would encourage the reader? This is just a suggestion to improve the quality of the paper.

- *We agree that the Introduction could be slightly trimmed at a few places and let appear a short paragraph mentioning the specific features reactivated such as sedimentary transport along the BGW, renewal of water in the wetlands and streams with consequences, for example, on "temperature refuges", or on retrieved biodiversity for fish populations, riparian woodlands, waterfowl, etc.*

- L162: "degraded the hydrological, geomorphological, and ecological functioning of the hydrosystem." This line and the following sentences would benefit from indicating a proper reference. Consider using 'impacted' instead of degraded if no detailed description (or reference) of those functioning can be given.

- *We agree that without clear evidences regarding the degradation, except relying upon multiple notes and reports mainly published in the grey literature (and hardly available), one could prefer to employ the notion of "impact" which is probably less negatively oriented.*

- Fig. 3: What is the base flow in this case ? How is it obtained ? It is not discussed in the main text.

- *The base flow is here associated with the continuous routine injection of 2 $m^3 s^{-1}$ in the new channel. As it is compared in Fig. 3 with injections of 70 $m^3 s^{-1}$, for visibility on the plot, base flow is marked with a red dashed line (just above level zero). The text associated with Fig. 3 could specify the value of 2 $m^3 s^{-1}$.*

- Fig 2 and 4 are not really the same… how have you decided to change the spatial structures ?

- *We agree that there exist differences in the zonation of hydrodynamic parameters compared with the spatial distribution of gravel bars in Fig. 2. Most differences appear as specific additional parameter zones along portions of the new channel and the BGW that are partly clogged (which is not witnessed by gravel bars) with delayed or smoothed responses of local subsurface head values to infiltration. Additional zones have been*

*delineated and parameters values have been adjusted during the calibration process to fit the local variation of heads in the subsurface.*

Fig. 5. KGE: I understand that you show both RMSE and KGE, because KGE does not change much when the simulation clearly matches less to the observation. This is probably due to compensations in the KGE terms and can be discussed in the main text by giving the three KGE terms (variance and bias ratio, and correlation coefficient).

- *We agree that the 3 KGE terms could be mentioned when the results are described. It will be done in the revised version*

- Fig. 5 and 6.: What are the boundary conditions time series on West and East sides ? Would it be relevant to show it? How much of the water comes from the side and how much from the flood? It could help to add up West and East bank boundary conditions and injection time series in this figure, for instance by lowering the size of the scatterplot.

- *In the context of both calibration and validation periods simulated in this study, the lateral (East and West) boundary conditions might slightly vary (as shown for instance by the measured groundwater level before peak injections in Figs 5 and 6), but we do not have enough data (water level measured each 15 days) to better condition boundary conditions. Therefore, it is useless to build an additional Figure. That being said, under routine injection in the new channel, the transverse (East-West) hydraulic head gradient in the Island is almost flat, very few water entering or exiting the system by the East and West boundaries. During peak injections, the increase in subsurface water levels inside the Island might change this relationship, even though groundwater head maps in Fig. 8 show that the main flow direction is still from South to North. One could if needed calculate the mean flow rates that escape through the East and West boundaries and mention the result in the manuscript.*

- L 297: "Results from particle size analysis also helped to predefine variation ranges of crucial parameters, such as the hydraulic conductivity and retention curve parameters of the sediments and the exchange coefficient between surface and subsurface.". This is key, have you any validation data of the calibrated values? Particularly over the different patches? Did you use pedotransfers functions? Which ones?

- *We do not have specific data, for example from infiltration experiments, permeameter tests, or well interference testing, to check on the relevance of the calibrated hydraulic parameter values. It is worth noting that such experiments could reveal not representative of parameter values at the scale of the zones that we employ to define the spatial distribution of parameters at the Island's scale. Nonetheless, when the piezometers (that are used for calibration and validation) were installed, soil cores were taken and analyzed in the lab to determine textural and granulometric characteristics at different depths and locations. We then relied upon the Rosetta model from US Salinity Lab (Riverside) to link textural properties of soils with main hydrodynamic parameters. The only validation that we can propose is to state that these calibrated values allow to fit heads (which is a poor validation given the well-known equifinalities on groundwater head distribution and transients resulting from the heterogeneity of a system, boundary conditions, etc.). The revised version of the paper could be amended to better explain this specific point.*

---

## Author Comment (AC2) · 9 Nov 2018

"Assessing the effect of flood restoration on surface-subsurface interactions in Rohrschollen Island 'Upper Rhine River – France) using integrated hydrological modeling and thermal infrared imaging", by B. Jeannot et al.

Reply to the comments raised by Rev #2
The discussion reported hereafter remind us with the comments of the Reviewer typed in straight font as our answers appear below in italic. In the reviewing process of HESS, a reply should be sent before being allowed to propose a revised manuscript. This is why some of our answers might appear as declarations of intend to which we would then try to stick in writing the revised version of the manuscript.
* * *
Jeannot et al. in the manuscript "Assessing the effect of flood restoration on surface subsurface interactions in Rohrschollen Island (Upper Rhine River – France) using integrated hydrological modelling and thermal infrared imaging" evaluate the efficiency of restoration actions adopted in Rohrschollen Island, in the Upper Rhine River specifically in terms of surface-subsurface flow exchange in the hyporheic zone. The surface subsurface interaction is quantified with a fully-distributed hydrological model, the Normally Integrated Model (NIM). The exfiltration areas are of great interest for this study, as they represent the opportunity for hyporheic exchange enhancement. Using an innovative approach, the output of the validated model in terms of exfiltration is compared with the information derived from thermal infrared imaging.

- *We thank the Reviewer for considering our approach innovative and for emphasizing the great interest associated with fine-scale modeling of surface-subsurface water exchanges in the context of hydrosystem restorations*

General comments
I appreciated reading this manuscript, I found it very instructive and I think it is a valuable contribution to HESS. Authors address a key issue for ecological restoration and suggest an innovative framework to evaluate the impact of anthropogenic activities. I have some suggestions I would like to see addressed in a revised manuscript. My first observation concerns the model setup. I might have misunderstood, but the surface-subsurface flow interaction model seems to be not considering some input/output terms related to the surface processes, e.g., the precipitation and evapotranspiration. If they are somehow included in the model parameters this should be pointed out in the manuscript and, if not, this hypothesis should be clearly stated and justified.

- *We thank the Reviewer for his (her) positive and encouraging appraisal. Regarding the boundary conditions at the surface of the Island, it is right that we neglected rainfall infiltrations and evapotranspiration (ET) processes. As we mentioned in another reply (to comments of Rev #1), the main reason to this simplification comes from the volume of water injected from the river to the Island, even under continuous base flow injections of 2 $m^3s^{-1}$. This flow rate mainly infiltrates in the subsurface through the bed of the injection channel and the BGW stream network. This base injection of 2 $m^3s^{-1}$ feeds the Island for a total amount of $6.3\times10^7$ $m^3s^{-1}$ in a year, which would be equivalent to 15 m annual rainfall infiltration over the whole Island. Even if only 20% of the injected volume reached the subsurface, this would still correspond to 3 m rainfall… For the short-terms events considered in this study and the forced flow conditions due to intense river water injections, one can consider that rainfall and ET are negligible. We agree that this rapid evaluation of injected volumes is not given in the manuscript. It could*

*help the reader to understand why rainfall and ET are neglected. We also agree that modeling long-term behavior, in the case of recessed river water feeding, should include hydro-meteorological forcing.*

 Another point related to the model setup concerns the time step and the time horizon of the simulation. These aspects are not discussed at all in the presentation of the model setup. Being this a study of the effects of some management policies, the reader expects a long-term problem setup. Moreover, the time step and the time horizon might influence the results and this should be discussed in the manuscript. In other words, the short time horizon implies calibrating the model on one event only, opening a debate about the robustness of the calibration, although the performance in validation might be convincing.

- *It must be acknowledged that the topic of this study is limited to short-term responses of the hydrosystem and the impact of artificial flash flooding on stimulated flow interactions between surface and subsurface compartments of the system. Evaluating quantitatively these impacts imposes modeling exercises with highly resolved models in time and space. In the present case dealing with a flat riverine island, high resolution in space allows for accurate delineation of wetlands and ponds that trap water in the surface compartment, and for accurate locations of infiltration versus exfiltration areas. The short time step employed goes with the capability of grasping highly transient behaviors including inversion of flow according to the contrasted dynamics between surface and subsurface compartments of the hydrosystem. That kind of simulation cannot be performed over large periods of time and is not designed for that. It is also noteworthy that Rohrschollen Island is a landscape recently restored (as many other examples in the World) which is still rapidly evolving due to the impacts of restoration works on the geometry of the system, its geomorphology, its land cover, etc. Another way to say that is to point out that the system cannot be simulated for long-term predictions on the basis of its current state. This notwithstanding, short-term simulations (as proposed in this study) and for various scenarios inheriting from information on the system over various past periods may serve so establish some probabilities (or statistics) on the responses of the system. For example, we show with our short-term simulations that flash flooding the Island with a peak injection of 80 $m^3s^{-1}$ fosters exfiltration over at least the next 45 days. Duplicating that kind of calculations for different periods in a year could allow to define the number and the dates of peak injections for maintaining a prescribed exfiltration water volume over a year. This will not address the robustness of the model for long-term predictions, but at least this gives some meaning and interest to short-term simulations when long-term forecasts and evaluations are requested. Perhaps slightly trimming the Introduction at a few places (as suggested by another reviewer) and adding at the end of the Introduction a short paragraph written in the sense evoked above could help to show that modeling accurately short events is not necessarily in complete opposition with long-term considerations on the modeled system.*

My second observation concerns the sensitivity of the results of the hydrological model to the calibrated parameters, such as the hydraulic conductivity. The authors claim that the exfiltration areas are the result of rapid infiltration which produces an important increase in groundwater level. However, the exfiltration areas do not always coincide with the temperature anomalies observed in the infrared imaging. The authors point out possible causes of the mismatch but do

not talk about the sensitivity of the model to the calibrated parameters. One could claim that the observed results might be due to an overestimation of the hydraulic conductivity in certain specific areas of the catchment. The discussion of this issue is required in the revised version of the manuscript.

- *Rev#2 is right when he suggests that there exist multiple reasons which could explain discrepancies between simulated exfiltration areas and temperature anomalies. We also agree that the model was not subjected to a complete sensitivity analysis which would assume, for robust results with non-linear processes, that the analytical differentiation of the state variables with respect to model parameters are calculated (the blunt method of perturbations is usually flawed or imprecise in these very transient and non-linear problems). Regarding over- or underestimations of hydraulic conductivity values, groundwater heads monitored over the Island are not very helpful to better condition conductivity, as they are strongly impacted by boundary conditions and water levels in streams…Nevertheless, the macroscopic hydraulic diffusion (the ratio of conductivity to specific storage) is correctly fitted as shown by the good match of observed heads both in time and amplitude. The point is that thermal anomalies are visible at a scale on the order of less than 10 m, which is also the scale of local heterogeneity of clay, sand, gravel, and pebble deposits in alluvial systems. A numerical model handling local heterogeneity at that scale should employ a mesh of 1-2 m sized elementary cells. In view of the available data, the mesh could not be assigned with a parameter value per cell, except in a stochastic framework generating various guesses on the parameter distributions and seeking the best ones. This is not the aim of this study which explores the possibility of simulating very transient behaviors of the hydrosystem and high contrasts of characteristic times associated with the flow processes. As is the case in many actual case studies, the model inherits from the few information available and is expected to render results at the large scale that roughly match up local data. Pinpoint accuracy is not expected. An additional paragraph could be added to better emphasize that :1- the measurement support of thermal anomalies is not that of the model , 2- flawed flow parameters could also explain discrepancies between simulations and observations, but at the scale of the Island, and given the hydraulic information available on the system, it is hard to obtain better results.*

Specific comments

L84-93: It would be interesting to have some examples of how increasing exchanges in the hyporheic zone contributes to the restoration projects.
L109: Wrong citation: "Fattichi" change into Fatichi.
L115: change computer into computational.
L124-126: Refer citations to the specific effect which is taken into account (water table dynamics, flood frequency, ecosystem services..).
L138-142: Better to express research objectives, possibly related to the discussion paragraphs: (1) model performance, (2) comparison between model results and TIR images and (3) comparison of different management options in terms of input quantity (and frequency, to add).

- *The five above comments could easily be accounted for in the revised manuscript by correcting typos, giving a few additional references, and rephrasing a few paragraphs.*

L296-299: The results from field experiments helped predefining the ranges of variations of some parameters of the model, but still some uncertainty exist on their calibration. It is worth

commenting on the uncertainty coming from the calibration of the parameters of the model here and/or in the discussion section. Moreover, some parameters that I assume were calibrated (e.g., the Manning coefficients) are not mentioned in Section 2.2.2. Maybe a table with the starting range of variation of all the calibrated parameters and the calibrated value could be useful.

- *We agree with Rev#2 that a table and a few comments on it could be helpful.*

L277-305: Here the simulation time step and time horizon should be discussed, because the reader does not know which is the reference time scale, also because restoration processes are usually associated with long time scales.

- *As told above when answering to a general comment on short-terms versus long-terms expectations regarding restored systems, we could add our view on the topic at the end of the Introduction. Having said that, a short reminder in due place (e.g., at the very beginning of Section 2.2.2) on the objective of the modeling exercise, the use of a highly resolved model run over short simulation periods, and its eventual usefulness for long-term predictions, could clarify the reading.*

L315: "After a first simulation employing the initial parametrization (defined in Section 2.2.2)" see comments below concerning the explanation of the model parametrization. It should be more exhaustive.

- *We agree that the paragraph is not fully clear and could be rewritten by stating that the calibration was performed by trial and error on the two types of sensitive model parameters (found by prior exploratory calculations) that are the hydraulic conductivity and the exchange coefficient. For the sake of simplicity, these parameters are spatially distributed as uniform values over subareas of the domain (zonation). The initial delineation of these subareas is consistent with the spatial distribution of sedimentary facies in Fig.2. However, when calibrating the model, a few zones have been added, especially along clogged portions of the BGW, to match local hydraulic head data (the only reliable hydraulic information available). The other parameters were prescribed at common values reported in the literature for similar environments. As preliminary calculations showed the weak sensitivity of these parameters, they were not calibrated…*

L320-322: "Only the hydraulic conductivity and the exchange coefficient between surface and subsurface were slightly adjusted while trying to preserve the initial spatial zonation" Not clear statement. Which is the initial spatial zonation? How was it defined? Does it mean hydraulic conductivity and the exchange coefficient are the only manually-calibrated parameters? What about the other parameters of the model? How were they fixed?

- *See our reply to the previous comment. We think that a better rewriting of the calibration procedure would shed light on the questions raised by Rev#2.*

L341-360: The discussion is consistent, but Figure 5 and Figure 6 are switched, so Figure 5 refers to the validation and Figure 6 to the calibration. This Section lacks of comments on the impact of the simulation horizon. The calibration on a single event does impact the results of calibration and validation. Please, comment on this.

- *Regarding Figs 5 and 6, this is an error when uploading the Figs for submission. Regarding the calibration and validation on two single events, we emphasized above that the aim of our study was not building a model for robust long-term predictions. The focus is here put on the feasibility of simulations highly resolved in time and space, probably the best way to decipher how flash floods impact at a fine scale over time and space the hydraulic behavior of the system. This focus could be reminded here. One could also remind that for systems under restoration processes and that still evolve in*

*their geometry, environment, and external forcing conditions, applying a current model (designed and calibrated on current information) over long-term horizons would miss that the system changed within the forecasting period.*

L354 and L357: It would be interesting to see the values of the three components of the KGE and their variation from calibration to validation.

- *This request was also pointed out by another reviewer and the three components could be given in the revised version.*

L371-377: Among this factors also the sensitivity to the model parameters should be pointed out. Somebody could claim that the observed dynamic might be due to an overestimation of the hydraulic conductivity implying higher infiltration and consequently exfiltration on a much larger area than the one where thermal anomalies are observed. Moreover, it is difficult to quantify the uncertainty related to the airborne TIR images, which were collected in a single survey. Maybe some comments on this uncertainty might help the reader to evaluate the results robustness.

- *As discussed earlier when dealing with the second main observation raised by Rev#2, model sensitivity to parameters is not straightforward to calculate in the present context. Having said that, discrepancies between simulated exfiltration areas and TIR images concern tiny spots that could be related to very local contrasts in hydraulic conductivity values. These small-scale contrasts cannot be mimicked in a model at the scale of the Island. Even though TIR images were carefully processed, one cannot overlook the fact that they represent a snapshot that could change within hours. We touch here the limitation of facing models with data that are not built (collected) on the same elementary support over time and space. In that case, we think that the robustness of the results is not in the fact that the model accurately represents data over a single scenario, but in the fact that the model roughly represents data over multiple different scenarios (events). Unfortunately, we only have at reach a single set of TIR. A comment in that sense could be added in the revised manuscript.*

L406-408: It could be helpful adding in Fig.10 also the injected flow reported in Fig.3.

- *This would probably overload the Fig, with a third scale for injected flow rates. But we could easily mark with bars along the time scale when the main injection peaks occur.*

L424: ".noting that the new channel was excavated in highly conductive sedimentary formations" this information comes from field experiments or from the parameters of the calibrated model?

- *This assertion comes from observations when excavating the channel mainly in pebble bars. This observation could be added.*

L464: "..but maintains increased areas of exfiltration over extensive periods" the definition "extensive periods" would have a more precise meaning if the problem in terms of time horizons was discussed while defining the boundaries of the problem. The observation is suitable also for "long periods" at L467. Section 3.4 is very interesting because it tests two different mitigation strategies in terms of input rate and volume, but under the title "Suggestions for management practices" some more information is expected, for example in terms of exchange frequency required over one year in order to observe ecological enhancement. Adding some information in this direction completes also the conclusions of Section 3.3, where the authors state "When forced injections enhance the development of wetlands and maintain high rates of exfiltration over long periods, from the mere hydrological standpoint, restoration works

are successful", but how often does it happen? How often should it happen in order to enhance the ecological status of the environment?

- *We agree with that, the notion of "extensive periods" or 'long-periods" being here defined with reference to the complete period over which a flash flood impacts the hydrosystem (a short-term event, in other words). We think that after clearly stating the motivation our study in the Introduction, and after reminding what is targeted when presenting the model settings, any reader should clearly grasp that the study is focused on short-events. In the same way as we also discussed earlier in this reply, it appears interesting to show how simulations of short-events as samples of the long-term behavior of the system may serve to partly foresee this behavior. We could had in Section 3.4 our view on the "statistics" (see above) provided by duplicating short-term simulations with the aim of guiding practices in relation with the long-term management of the system (as the example proposed above defining the number of flood events needed to maintain a prescribed volume of exfiltration in a year).*

Figure 10: Add the pattern of the inflow. Right y-axes change m3/s into m3/s. If possible, it would be nice to have enhanced image quality of Figure 5, 6, 8, 9 and 11.

- *See above, bars along the time scale would provide the dates of the peak injections without overloading Fig. 10. Regarding Figs in general, the quality of the originals is far better than the low-resolution pdf format proposed by the Journal for reviews. The original quality seems to be enough for further eventual electronic publication.*

---

## Author Response (AR2)

Comments to the Author:

Dear authors,
thank you for submitting the revised version of your manuscript. Considering the changes and responses to the referee's comments that you have provided, I have the pleasure to inform you that our contribution can now be further processed towards publication in HESS. Prior to that, I would ask you to carry out a few technical corrections. Please consider the points here below (by simple comparison to the text in your revised manuscript):

465 The lack of data suggests that perfect accuracy cannot be expected, …

475: Modified in the revised version

466 … between the size of measurement supports and model resolution is the main reason for What does 'measurement supports' exactly mean here ?

476: "Measurment supports" was used here for measurement resolution. It is changed in the revised version.

468 …, we suggest that the quality of model results does not relate to the fact that the model accurately represents data over a single scenario, but rather to the fact that it roughly represents data over multiple different scenarios (events). Unfortunately, we only had one single set of TIR imagery at our river reach.

478-481: Modified in the revised version.

563 As already mentioned, the short-term behavior of the hydrosystem in response to flood events

576: Modified in the revised version.

566 The exploration of injection scenarios …

579: Modified in the revised version.

568 … simulations, could for example inform on the number and intensity of flood events needed to …

581: Modified in the revised version.

I am looking forward to see your manuscript published in HESS. Thank you for considering our journal for publishing your research results.

Best regards,

Laurent Pfister